# Algal Consortiums: A Novel and Integrated Approach for Wastewater Treatment

Prateek Gururani [1], Pooja Bhatnagar [2], Vinod Kumar [2,3,*], Mikhail S. Vlaskin [4] and Anatoly V. Grigorenko [4]

1 Department of Biotechnology, Graphic Era (Deemed to be University), Dehradun 248002, Uttarakhand, India
2 Algal Research and Bioenergy Lab, Department of Life Sciences, Graphic Era (Deemed to be University), Dehradun 248002, Uttarakhand, India
3 Peoples' Friendship University of Russia (RUDN University), Moscow 117198, Russia
4 Joint Institute for High Temperatures of the Russian Academy of Sciences, 13/2 Izhorskaya St., Moscow 125412, Russia
* Correspondence: kumarvinod.ls@geu.ac.in

**Abstract:** Urbanization, industrialization and other human-related activities discharge various inorganic and organic toxic compounds into the environment. Many physical, chemical and biological methods have been practiced, to treat contaminated wastewater: among these, the biological method of wastewater treatment by utilizing algae has been reviewed widely. However, the removal efficacy of algae monoculture is low, as compared to the algae consortium systems. The presence of microorganisms such as fungi or bacteria in wastewater can establish various relationships, such as mutualism or symbiosis with algae, which help in the removal of various organic and inorganic compounds from wastewater, thus acting as a wastewater treatment system. Heterotrophic microorganisms can segregate natural organic matter, which is released by algae in the form of dissolved organic carbon, and releases carbon dioxide, which is utilized by algae for photosynthesis. In accordance with existing studies, microalgal consortiums with bacteria or fungi occurring naturally or crafted artificially can be utilized for wastewater treatment; therefore, the present review provides an outline of the symbiotic relationships between algae and other microorganisms, and their applications in wastewater treatment. Various mechanisms—such as mutualism, commensalism and parasitism—for the removal of different pollutants from wastewater by consortium systems have been elucidated in this review; moreover, this review addresses the challenges that are restricting large-scale implementation of these consortiums, thus demanding more research to enable enhanced commercialization.

**Keywords:** wastewater treatment; algae; algae–bacteria consortium; algae–fungi consortium

## 1. Introduction

Water, along with air, is the most precious and liberal resource for human survival [1]; however, in recent decades, the constant development of societies, and their increased dependence on fresh water sources, have led to the extensive generation of wastewater from different non-pointed and pointed sources, such as food wastewater, industrial wastewater, domestic wastewater and many more [2]. Wastewater constitutes various contaminants and pollutants, involving nutrients such as phosphorus and nitrogen, and heavy metals such as lead and zinc, which are of emerging concern; furthermore, it has been reported that in 2020, emissions of total phosphorus and total nitrogen reached 336,700 tons and 3,223,400 tons, respectively. In addition, emissions of chemical oxygen demand (COD) were five times greater in 2020 than in 2019, extending to 25.6476 MT, and the overall discharge of heavy metals reached around 26,680 kg [3]. If the wastewaters are directly released into the environment without any effective treatment, such toxic pollutants will not only harm aquatic life, but will also risk human health [4]. It is estimated that by 2030 the world will be faced with a 40% water shortage in existing water resources such as rivers, lakes and

glaciers [5]. In light of the above-mentioned problems, development of an efficient and sustainable wastewater treatment system is of the utmost importance.

Presently, the major wastewater treatment methods include physical methods (filtration screening, sedimentation, membrane filtration), chemical methods (ion exchange, chemical precipitation, electrochemical treatment, adsorption) and biological methods (bioprecipitation, biosorption, biological activated sludge) [2,6]. The positives of these methods for treating wastewater are very well known, but the methods are also associated with certain limitations, such as irregular removal efficacy, and the uneconomical and increased cost of installation, which can increase the problem of successive treatment, resulting in secondary pollution [7]. Similarly, biological methods are related to the problems of kinetics, maintenance of a favorable environment and the low biodegradability of some pollutants. As compared to bioprecipitation and biosorption, activated sludge is a more beneficial method, enabling high removal of suspended solids and biochemical oxygen demand: however, there are also issues of poor decolorization and the possibility of sludge foaming and bulking [8]; therefore, scientists are searching for more effective and sustainable methods of treating wastewater—in which regard, microalgae have received increased attention. "Microalgae" is a common term that is generally used to describe photosynthetic microorganisms, such as prokaryotic cyanobacteria and eukaryotic microalgae [9]. Microalgae and wastewater treatment have been connected to one other from ancient times [10,11]. Oswald and Gotaas [12] initiated the utilization of algae to decontaminate sewage in the 1950s, which opened the doors for incorporating algae into wastewater treatment: this was because microalgae exhibit a more remarkable ability to consume nutrients, which accounts for around 80–100% for phosphorus and nitrogen, offers a high carbon fixation rate, and can also solve the energy-related problem [13–17]. More recently, it has been stated that the co-cultivation of microalgae with other microorganisms, which are either present naturally in their growth media or added, is a more promising approach, which could assist the process of cell division, in addition to producing an extensive variety of metabolites, which would have great economic significance [18]: this is because the integration of microorganisms into various metabolic activities permits the development of a powerful biological system, which can function under varying nutrient loads and environmental conditions [19–21]. The symbiotic relationship between algae and other microorganisms was first outlined in the early 1950s, in the process of boosting the supply of oxygen in oxidation ponds at wastewater treatment plants [18]. Moreover, collaborative relationships can be developed among the microorganisms by incorporating consortiums that can enhance the rate of nutrient removal [9]. Existing studies have demonstrated that the presence of microorganisms such as fungi or bacteria in algae cultures can initiate a positive influence in algal cell growth [22,23].

The aim of this review is therefore to provide a deep understanding of the symbiotic relationships established between microalgae and other microorganisms in wastewater treatments, including the mechanism for the removal of various contaminants and pollutants from wastewater. The paper further addresses the challenges that are restricting large-scale implementation of these consortiums, thus demanding more research and effort, to enable enhanced commercialization.

## 2. Wastewater and Associated Conventional Methods for Its Treatment

The phrase "wastewater" can be defined as "any water whose chemical, physical or biological composition has been changed as a result of direct discharge of multiple pollutants into water bodies either from urbanization, agricultural, industrial or domestic activities hence making it unsuitable for potable and other purposes" [1]. Generally, wastewater may constitute huge amounts of inorganic compounds, organic pollutants, sediments, pathogenic microorganisms, oxygen demanding wastes and nutrients like phosphorus and nitrogen [24]. In addition, the composition of wastewater is strongly influenced by its sources: for instance, wastewater discharged from the swine industry represent increased phosphorus and nitrogen levels, as compared to municipal wastewater, while wastewater

coming from dairy, starch, brewery or potato-processing industries reflects an enhanced proportion of soluble chemical oxygen demand [15]. However, this speedy generation of wastewater from multiple non-pointed (agricultural and urban run-off) and pointed (industrial effluents, municipal sewage, combined sewer overflows) sources contaminates our environment, and is responsible for inducing certain unacceptable changes in aquatic habitat, ultimately leading to harmful effects such as the occurrence of water-borne diseases (diarrhea, typhoid), shortage of drinking water, extinction of aquatic life, contamination of freshwater sources, and many more [23,25,26], as shown in Figure 1.

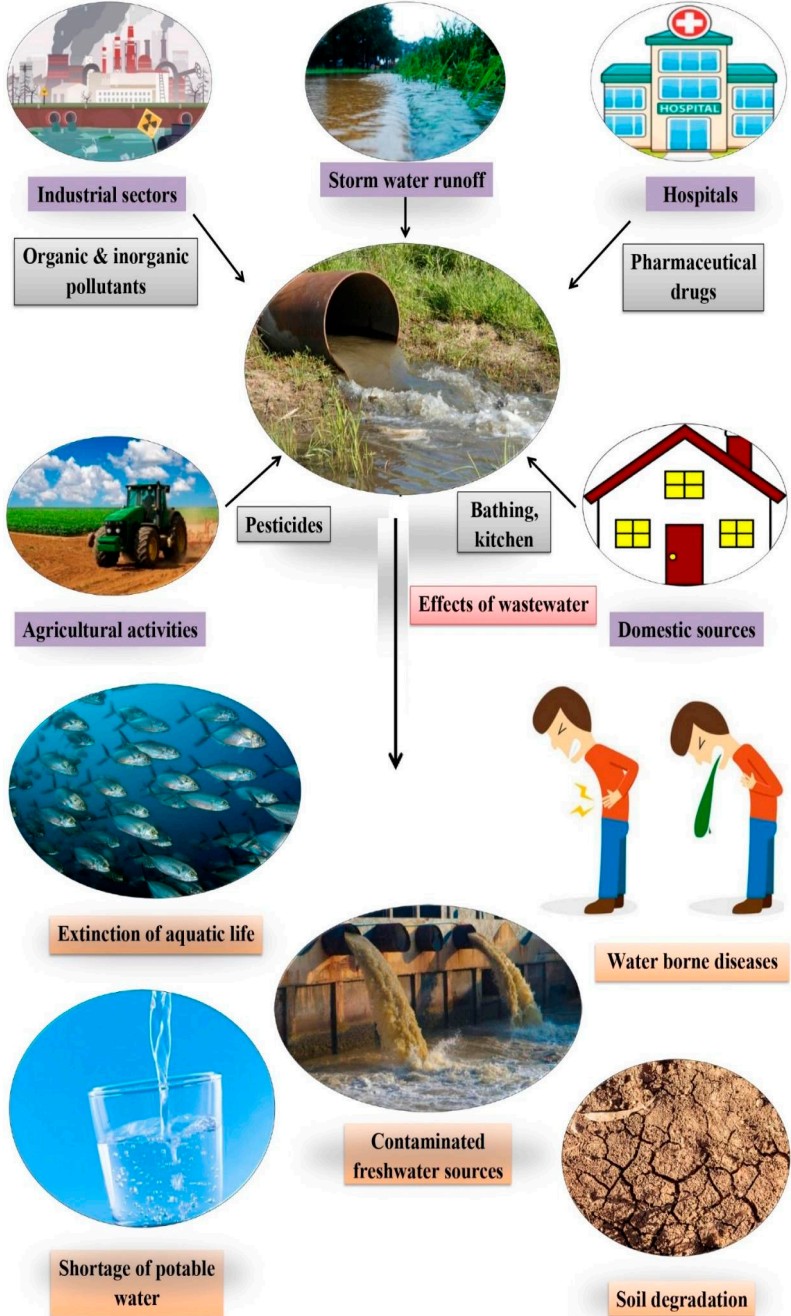

**Figure 1.** Different sources of wastewater, and its adverse effects on the environment and society.

## 3. Microalgae and Their Cultivation

Microalgae are defined as a prevalent class of oxygen-producing photosynthetic organisms, analogous to plants, which extensively survive in multiple water environments,

including marine and freshwater and a diversity of wastewaters, such as industrial, agricultural, municipal and many more. Microalgae possess a great variety of industrial applications along with biological importance, such as carbon sequestration, photosynthesis thus producing oxygen, and the utilization of nutrients such as nitrogen and phosphorus from wastewater [27–29]; therefore, the coupling of microalgae cultivation with wastewater treatment can be seen as a promising approach to growing algae, accompanying the wastewater treatment process [30]. Importantly, microalgal cultivation is supported by the presence of an extreme concentration of nutrients (nitrogen, phosphorus) in the wastewater, as high quantities of carbon will result in a faster growth rate [31,32]; moreover, light intensity, light quality and photo-bioreactors (open or closed) promote algal cultivation [33]. The main requirement of light in algal cultivation is for carbon fixation, enhancing the growth rate of the algae [34,35]. Microalgae may possess various kinds of metabolism, including autotrophic, mixotrophic and heterotrophic [36], as shown in Figure 2; therefore, selecting an appropriate cultivation system for microalgae is a principal step towards influencing the algal growth rate and the efficiency of the desired process.

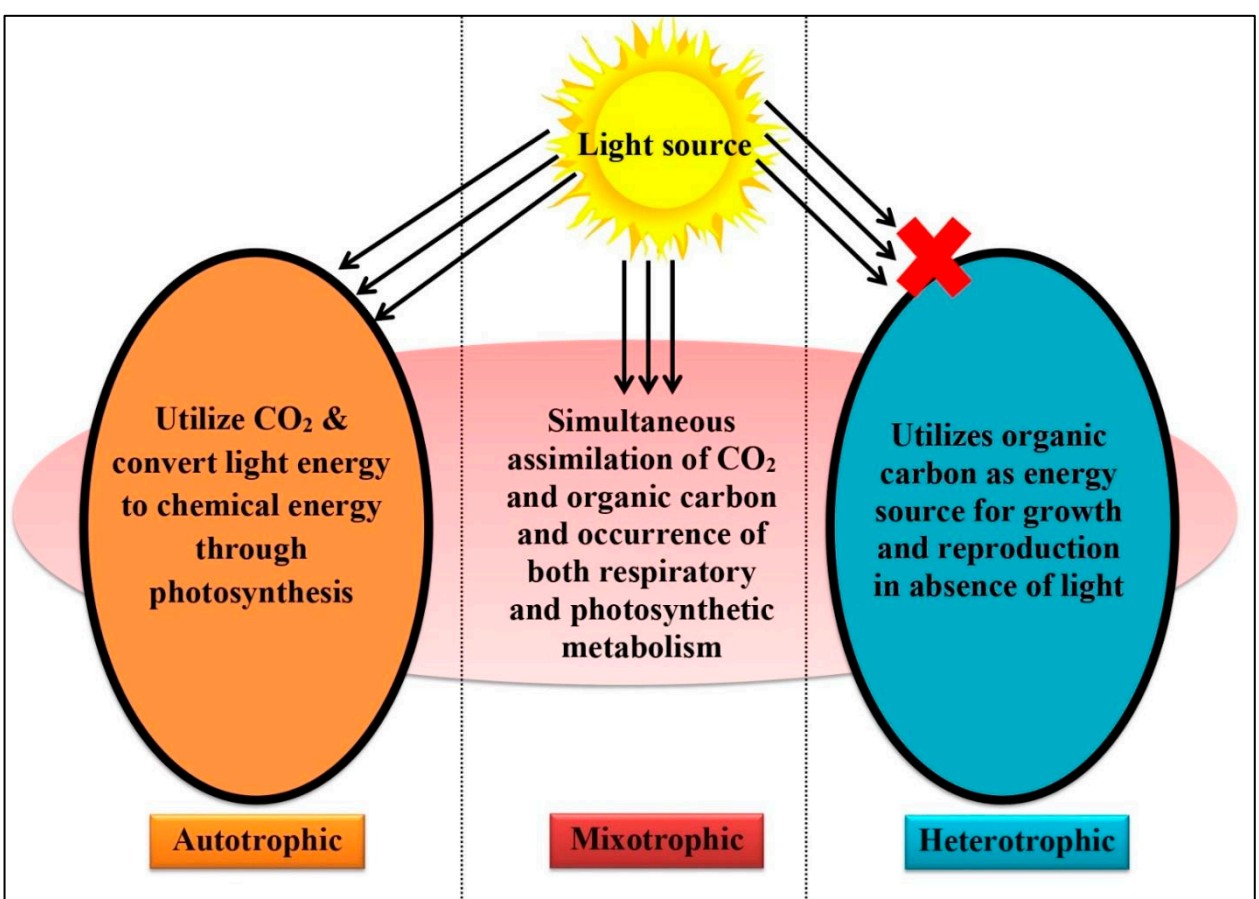

**Figure 2.** Different types of microalgae metabolism.

On the basis of the design conditions, there are two types of cultivation systems: open and closed pond systems, as shown in Figure 3. In an open system, algae are generally cultivated in open area surroundings, such as scrub, deep channels, tanks, lagoons, shallow circulating units and raceway ponds [37,38]. In this type of system, nutrition and water are provided to microalgae by channeling runoff water from neighboring water treatment plants, industrial disposal water or land areas [32]. The most commonly used type of open pond system is the raceway pond, because of its efficiency in generating a high amount of microalgae for economic application. The major drawback of raceway pond systems is that they are an obstacle to controlling the surrounding environment conditions, such as

weather and temperature, which possess a direct influence on biomass productivity [39]. To overcome the issues associated with open pond cultivation systems, photo-bioreactor (PBR) technologies have been designed, in which algae are cultivated in vessels with transparent walls, and are exposed to artificial light, thus enabling photosynthesis. PBR allows the cultivation of microalgae for a longer period, as compared to open pond systems, hence producing a high amount of algal biomass [32,38]. There are multiple types of closed photo-bioreactor systems, such as the flat plate type, the tubular type and the column type of photo-bioreactors, which are more productive and practical for algal cultivation, because of their efficiency in significantly controlling the surrounding environment conditions, such as temperature, pH and $CO_2$ concentration, and further reducing the chances of contamination [32]. The increased cost of maintenance and construction is a major challenge associated with closed photo-bioreactor systems: however, few studies have communicated that this high cost can be minimized by utilizing wastewater as a growth medium, employing cheap and efficient materials and energy-effective pumps [40].

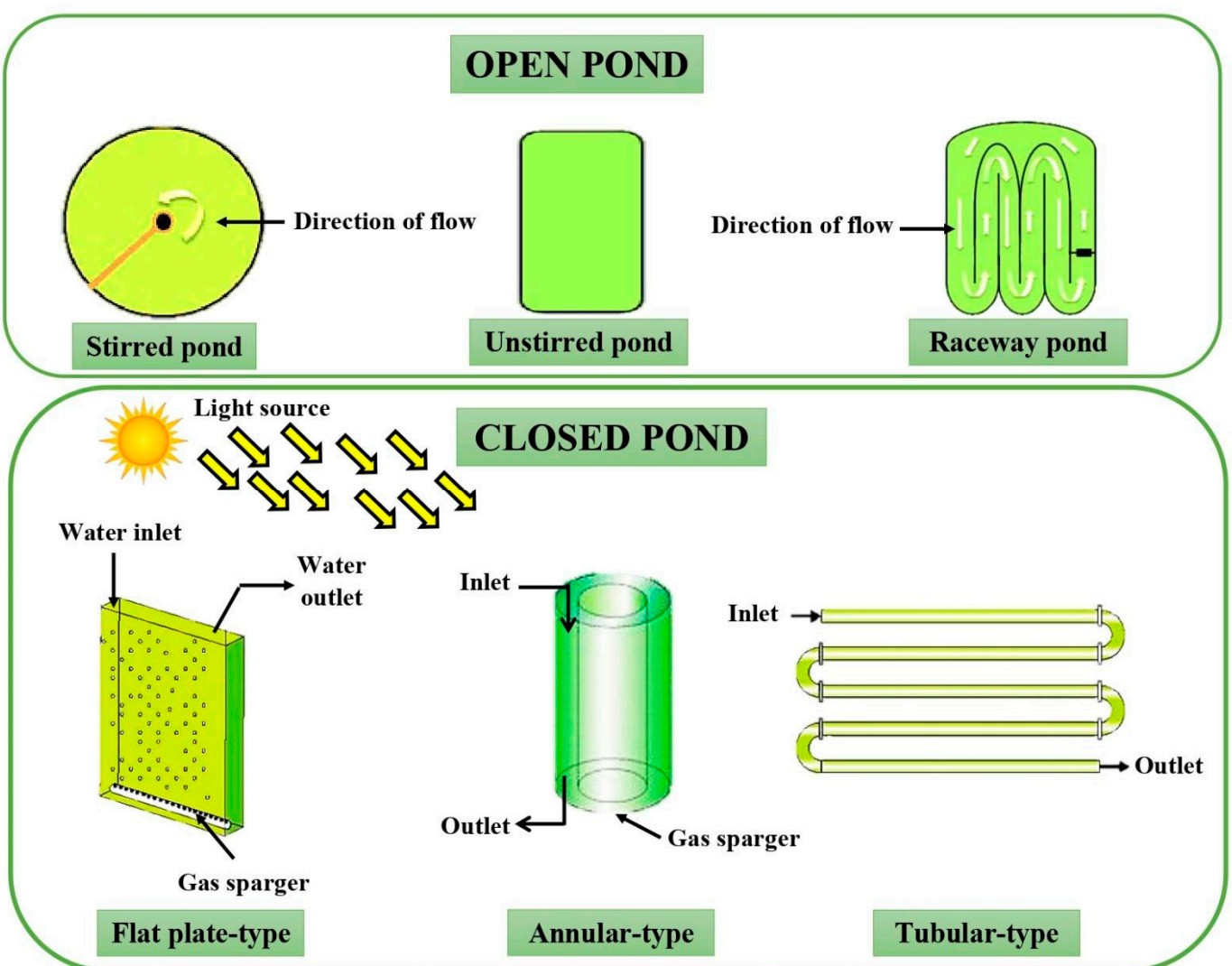

**Figure 3.** Open and closed pond cultivation system of microalgae.

## 4. Working Action of Algae and Their Different Consortiums for Wastewater Treatment

Although microalgae have been effectively utilized in nutrient removal from various wastewaters, the maintenance of microalgae monoculture in such processes is quite challenging; therefore, some of the existing studies have communicated the benefits of utilizing microalgae consortiums above single-species cultures such as *Chlorella vulgaris* [41,42],

*Scenedesmus obliquus* [41] and *Halochlorella rubescens* [15,43–47]. For instance, the complicated process involved in the breakdown of various pollutants might be difficult to achieve with monocultures: however, there would be benefit in the utilization of microalgae consortiums. In addition, the implementation of such consortiums could lead to the emergence of a powerful system that would be capable of resisting interruption by other species and varying environmental conditions [15,45]. Such consortiums can occur naturally in the environment [48]: for instance, in various types of wastewater, such as landfill leachate, agricultural, domestic, municipal or industrial wastewater [49–51]. Moreover, the existence of other microorganisms, such as bacteria or fungi, in wastewater represents a vital role in boosting microalgae growth and nutrient removal [15]. Furthermore, consortiums can be artificially engineered, through the association of microorganisms which do not naturally co-exist, for a particular purpose [48]. These consortiums include: the association of one microalga with another (algal–algal consortium)—for example, *Chlorella, Scenedesmus, Chaetophora* and *Navicula*; bacteria (microalgae–bacterial consortium)—for example, *Scenedesmus obliquus* and *Bacillus megaterium*; and fungi (microalgae–fungi consortium)—for example, *Chlorella pyrenoidosa* and Rhodosporidium toruloides [15,22]. The following section describes the different types of associations which can be implemented among microorganisms by incorporating these consortiums, and how such relationships can enhance the efficiency of wastewater treatment.

*4.1. Working Action of Algae–Algae Consortiums for Wastewater Treatment*

In the interactions between photosynthetic organisms, it has been assumed that cultivating such organisms in a consortium could lead to both competitive and cooperative associations: on the one hand, these microorganisms might display cooperative associations by exchanging metabolites, leading to the final enhancement of biomass productivity, and thus increasing the efficiency of nutrient removal [52]; however, co-cultivation of photosynthetic organisms could lead to the discharge of secondary metabolites, also known as allelochemicals, which reveal an adverse influence on the co-cultivated microorganisms [15]. The production of allelochemicals can be suppressed or enhanced by both biotic and abiotic factors. Nutrient starvation, increased pH, low temperature and light intensities are the primary abiotic factors which can increase the production of allelochemicals, whereas extreme supply of nutrients, low pH, increased temperature and light intensities may restrict the production of allelochemicals; moreover, the biotic factors that affect the production of these secondary metabolites include the concentrations of the microorganisms involved [52]. Such interactions among photosynthetic microorganisms have several benefits for wastewater treatment processes: firstly, they boost the utilization of complete nutrients, if the nutrients are supplied in an adequate amount; secondly, they withstand predators and contaminants, by initiating the production of allelochemicals; thirdly, there is an establishment of a settleable system achieved by the combination of a single cell organism with flocculating ones, hence excluding the necessity of a harvesting method [9]. In particular, various microalgae species, such as *Chlorella sorokiniana*, *Chlorella vulgaris*, *Tetradesmus* sp., *Ascomycota* sp., *Chlorella saccharophila*, *Chlamydomonas pseudococcum*, *Scenedesmus* sp., *Neochloris oleoabundans* and *Coelastrum microporum*, have been utilized for treating wastewater coming from different sources, such as meat processing wastewater, tannery wastewater, dairy wastewater and activated sludge [53–56]. In addition, the utilization of microalgae consortiums in wastewater treatment guarantees the feasibility of the decontamination process. This is because the loss of one microorganism could be equilibrated by some other microorganisms incorporated in a consortium [9]. Figure 4 highlights the mechanism involved in the removal of nutrients by microalgae.

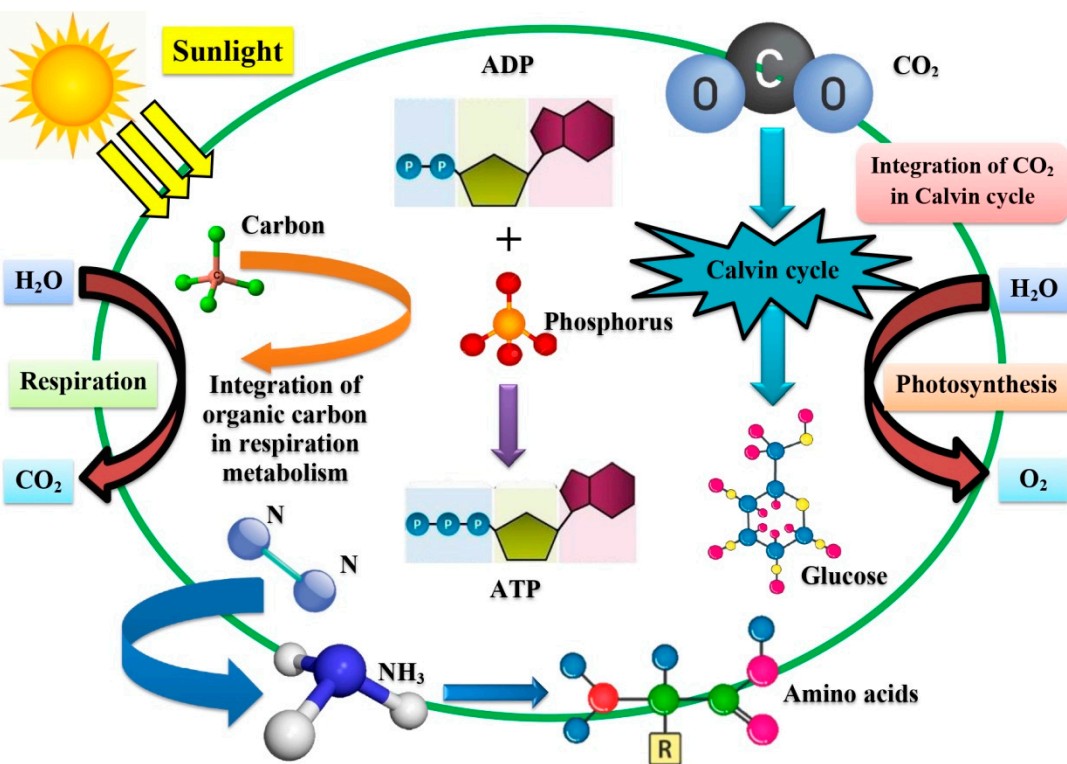

**Figure 4.** Nutrient removal in wastewater by microalgae.

*4.2. Working Action of Algal–Bacterial Consortium for Wastewater Treatment*

Microalgae and bacteria develop a complex symbiotic relationship that is utilized in the process of wastewater treatment [57]. On the one hand, bacterial species can degrade the organic content, such as carbohydrates, proteins, fats, oils, pesticides and phenols, present in wastewater into water and carbon dioxide, and the carbon dioxide produced by the degradation turns into a carbon source for the microalgae, thus promoting its photosynthesis [58]; in addition, the metabolites of the bacteria can be transported into the cytoplasm of the microalgae [59]. On the other hand, microalgae can enhance the metabolic potential of bacteria or similar microorganisms by generating oxygen through photosynthesis, which decreases the oxygen aeration filling costs, as there is natural production of oxygen, hence saving energy and minimizing the consumption in reactors [58]. Moreover, nutrients like nitrogen and phosphorus can be utilized by microalgae–bacterial consortiums, thus improving wastewater quality [59]. A consortium can efficiently fix the microalgae, thus minimizing its loss, and potentially can settle biomass at the time of outflow. As compared to physical, chemical and electrical methods, cultured ubiquitous microorganisms, such as self-flocculating algae, fungi, bacteria and yeasts, are more efficient and chemical-free materials for gathering target algal strains through bioflocculation [60]. Physical methods like centrifugation, floatation and filtration can attain high efficiencies, but the operational costs are too high. Gravity sedimentation can save energy, but its application is limited by species-specific and time consumption features. Negatively charged algae can further be concentrated by electrical methods, but the development of electric fields demands huge capital expenditure. Similarly, chemical methods are associated with the problems of biomass contamination, due to the involvement of complicated chemical reagents [61]. Therefore, co-culturing of algae with other microorganisms helps to immobilize microalgae and, under certain cultivation conditions, algal cells can develop spherical morphology, with various benefits such as improved mass transfer rate, high mechanical stability and large surface area. More importantly, the cell pellets can be separated from the culture broth through a sieve, due to their large size, thus reducing operational costs [22]. Furthermore, while absorbing phosphorus, nitrogen, carbon and similar nutrients in wastewater, microal-

gae can produce polysaccharides, proteins, oils and related compounds in wastewater, that can be utilized as bioenergy to tackle energy issues in upcoming years [62]. However, the generation of biofuels is accompanied by difficulties, such as the recovery of many soluble catalysts from the end products, which is challenging, requiring intensive energy and costly separation technologies. Similarly, the solvent needs to be recovered either by evaporation or by distillation whereby, along with the solvent, small molecules of bio-oil can also be lost, thus resulting in reduced bio-oil yield [63]. Existing studies have reported on the possible efficiency of a microalgae–bacterial consortium system involving several species—such as *Tetraselmis indica* and *Pseudomonas aeruginosa*; *Scenedesmus obliquus* and *Bacillus megaterium*; *Chlorella prototothecoides* and *Brevundimonas diminuta*; and *Chlorella vulgaris* and *Exiguobacterium*—to treat various wastewaters, such as dairy wastewater, biogas slurry, and piggery wastewater [64–67]. Moreover, the symbiotic relationship between bacteria and algae can be elucidated with three different types of associations: mutualism, commensalism and parasitism. Algae can efficiently make use of nutrients that are available in wastewater as a source for producing renewable energy. In addition, the microorganisms related to wastewater interact indirectly or directly with the microalgae through any of the above-mentioned interactions, which can result in hindrance or betterment of the species involved [68].

Bacteria and algae are the decomposers and producers of the ecosystems in which they reside [69]. Heterotrophic bacteria are well known for decomposition, and in consortiums with microalgae they can establish mutualistic association [59]. The mutual relationship between bacteria and algae can be categorized into four classes: nutrient exchange, nitrogen fixation, gene transfer and signal transduction [70]. The initiation of a mutual relationship is advantageous for promoting growth and nutrient transmission between bacteria and microalgae. Furthermore, the mutualistic surroundings established by bacteria and microalgae are an important influence on the biological treatment of wastewater [58]. Existing studies have revealed that micronutrients involving macro-elements—such as carbon and nitrogen, vitamins and plant hormones—are interchanged among bacteria and algae. In symbiotic associations, algae supply organic carbon for the symbiotic bacteria [71] and, in return, the bacteria supply low molecular weight organic carbon and inorganic carbon for the microalgae [72]. Moreover, regarding the algal growth, bacteria can decompose organic matter and mineralize it, thus boosting its growth rate, and further supplying minerals to the microalgae [73]. The algal growth is influenced by the amount of phosphorus and nitrogen, sulfur and carbon present in the environment: if there is a lack of these or similar elements for a long period of time, the growth of the microalgae will stop, or the cells will experience apoptosis. Contrastingly, the growth of microalgae in nutrient-rich wastewater can result in the accumulation of algal blooms in huge amounts, which can be toxic to aquatic life [58]: on that account, heterotrophic bacteria are cultivated along with algae, because they consume carbon sources and other nutrients, thus preventing bloom formation (a type of mutualism), as shown in Figure 5. Therefore, developing adequate mutualistic relationships is essential among bacteria and algae for the good treatment of wastewater, which can further enhance the biochemical activities of microalgae and bacteria, which in turn is very helpful for the synthesis of algal biomass [62]. Moreover, existing studies have noted that bacteria can further promote self-aggregation in microalgae, which is immensely beneficial in the engineering field [58].

If the nutrients available in the environment are unable to fulfill the requirements of bacteria and algae in a mutual association, then commensalism takes place between the two [74]. Commensalism is a biological interaction whereby only one partner benefits from the interaction: for example, *Chlamydomonas reinhardtii* utilizes bacteria-delivered vitamin B12, whereas the bacteria do not use the algae's carbon [23]. Bacteria and algae develop surroundings more appropriate for the survival and expansion of their own communities, and competitive association thus occurs between the algae and the bacteria: for example, existing studies have revealed that when bacteria are cultivated in reduced phosphate content, then the bacteria compete with the algae for the phosphate, and the bacteria utilize the phosphate more effectively than do the algae [73], whereas in cases of reduced

nitrogen content, the algae compete with the bacteria, and reveal a higher growth rate [58]. Sometimes bacteria can also act as parasites, thus affecting algal growth: for example, enzymes like chitinase, glucosidase and cellulose can disintegrate microalgae cells, due to which the intracellular components of the algae are utilized by the bacteria as a source of nutrients [75]. It is also notable that healthy algal cells possess the ability to restrain the colonization of bacteria on their surfaces, and that they might be hindering the uncontrolled growth of bacteria, decreasing the availability of light and nutrients. Furthermore, the biofilm of bacteria can destroy algae, as bacteria possess the capability to penetrate inside the algal tissues, giving rise to diseases [76].

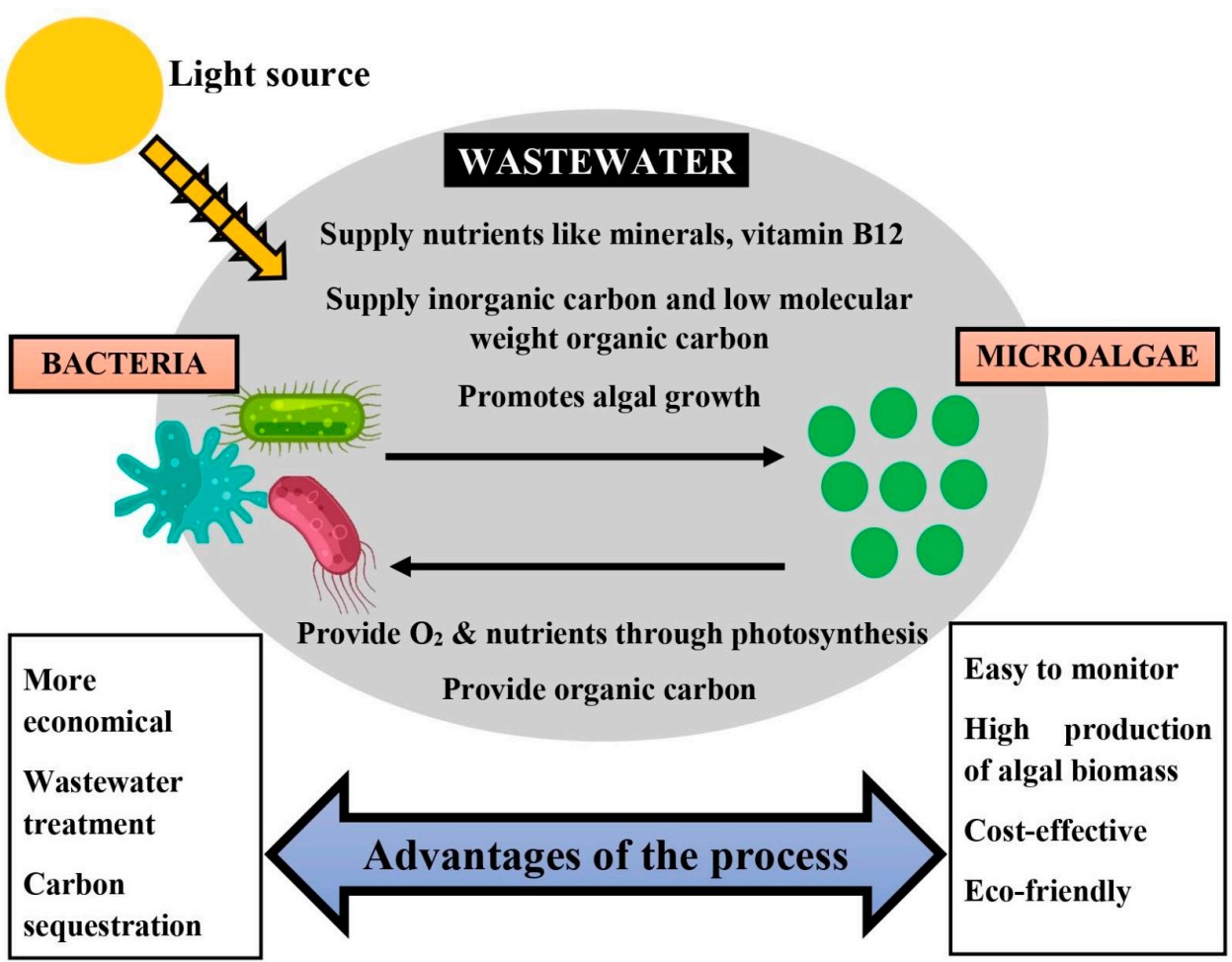

**Figure 5.** Symbiotic relationship between microalgae and bacteria.

### 4.3. Working Action of Algal–Fungi Consortium for Wastewater Treatment

Fungi are the heterotrophic organisms which can transform organic contents into carbon dioxide by metabolism, whereas inorganic carbon sources are utilized as raw materials by autotrophic microalgae for biomass accumulation [77]; hence, the oxygen released by microalgae during photosynthesis can be utilized completely by fungi for respiration, providing carbon dioxide back to the microalgae cells. Apart from decreasing the concentration of nutrients, the utilization and transfer of extensive carbon sources in the habitat of wastewater further stimulates the accumulation of biomass for value-added compounds involving protein-rich feed, biogas and biodiesel [78–80]. Furthermore, some nutrients—mainly carbon and nitrogen—are deep-seated in suspended matter, thus creating a difficulty for microalgae in utilizing them directly [22]. However, in co-cultivation conditions, macromolecular organic content can be transformed into soluble low-molecular-weight nutrients along with the action of extracellular enzymes of fungi: therefore, microalgae can poten-

tially eliminate multiple nutrients from wastewater by assimilating the enzyme-treated soluble contents [81]. In particular, because of the mutual reinforcing mechanisms established between fungi and microalgae, this co-cultivation method can be more efficacious for eliminating various nutrients—such as chemical oxygen demand (COD), phosphorus and nitrogen—from wastewater, as compared to a monosystem [22]. Specifically, different species of microalgae and fungi—such as *Chlorella pyrenoidosa* and *Rhodosporidium toruloides*; *Chlorella vulgaris* and *Aspergillus* sp.; and *Scenedesmus* sp. and *Trichoderma reesei*—have been utilized successfully in treating several wastewaters, such as distillery, domestic, swine manure wastewater and secondary effluent [78,82,83].

Some heavy metals, including cobalt, zinc, manganese and copper, are crucial for the growth of fungi and microalgae as trace components, and are further engaged in the cell metabolism and enzymatic process, whereas other heavy metals—such as mercury, cadmium, arsenic, chromium and lead—are harmful to the organisms [49,84–86]. In recent years, co-cultivation of fungi and microalgae has been regarded as a powerful approach to the treatment of wastewater contaminated with heavy metals. The utilization of algal–fungi consortiums for the biodegradation of wastewater constituting heavy metal ions takes place in two stages. Initially, there is speedy extracellular passive adsorption of the metal ions on the cell surface, by a number of mechanisms including surface complexation, physical adsorption, ion exchange and micro-precipitation [87,88]. The cell wall of fungi and algae basically comprises proteins, lipids and polysaccharides that can assist huge metal-binding functional groups such as hydroxyl, amino, phosphoryl and carboxyl [89]; moreover, the atoms of oxygen, sulfur, phosphorus and nitrogen in functional groups can supply heavy metal ions with an unshared pair of electrons that are complex and coordinate, thus ensuring the secure bonding of heavy metals to the cell wall [90]. In the second stage, there is agglomeration of heavy metal ions inside the cell, which is slower than the first stage, as the method is an energy-driven metabolism: following the adsorption of heavy metals on the cell surface, they are actively transferred into the cytoplasm by cell membrane, and link with the internal binding sites of peptides or proteins [91]. Moreover, the cell organelles, such as mitochondria, vacuoles and chloroplasts, initiate the combination of heavy metal ions with organic molecules like sugar, sulfide and protein, resulting in complex formation; hence, heavy metal ions are accumulated in the form of polyphosphates or sulfides within the cells [86], as shown in Figure 6.

The remediation of large molecular organic pollutants like pesticides, pharmaceuticals, detergents and petro-alkane through microalgae–fungi consortiums, generally involves three mechanisms: (i) bio-adsorption, (ii) bio-uptake and (iii) biodegradation. The methods of bio-uptake and bio-adsorption are identical, as in the case of bioremediation of heavy metals [22]: however, a difference exists in the degradation of the pollutants within the cells, i.e., the organic pollutants can break down into small molecules by going through a sequence of biochemical responses while, on the other hand, heavy metals cannot be degraded within the cells [92]. Moreover, nutrients like phosphorus, carbon dioxide, nitrogen and organic carbon are crucial for the growth and photosynthesis of algae in a consortium system [93,94]: hence, for simultaneous remediation of nutrients, an extremely practicable microalgae–fungi consortium technique must be utilized. In wastewater, a high amount of freely available nitrogen is basically accessible in the form of ammonia, nitrates and nitrites that possess a very crucial part in the metabolism through assimilation [14]. Nitrogen is required for the production of proteins, nucleic acids and phospholipids [95]. Phosphorus is required for the production of adenosine triphosphate (ATP), lipids and nucleic acids [96,97]. Microalgae are recognized as autotrophic organisms, which demand nitrogen to produce proteins, nucleic acids and phospholipids [95]. In addition, phosphorus is one more macronutrient that is also essential for the production of the adenosine triphosphate (ATP), lipids and nucleic acids of the cells [96,97]. Various forms of inorganic phosphates, such as $H_2PO_4^-$, $PO_4^{3-}$ and $HPO_4^{2-}$, are engaged in the production of organic elements through phosphorylation, resulting in an increased potential for nutrient removal in wastewater [14]. Earlier studies have demonstrated that, compared to microal-

gae, fungi have a vastly superior ability to eliminate chemical oxygen demand (COD) from wastewater [98]: this may be because fungi are heterotrophic organisms that utilize organic carbon as their single carbon source and main energy type, leading to efficient depletion of COD [78].

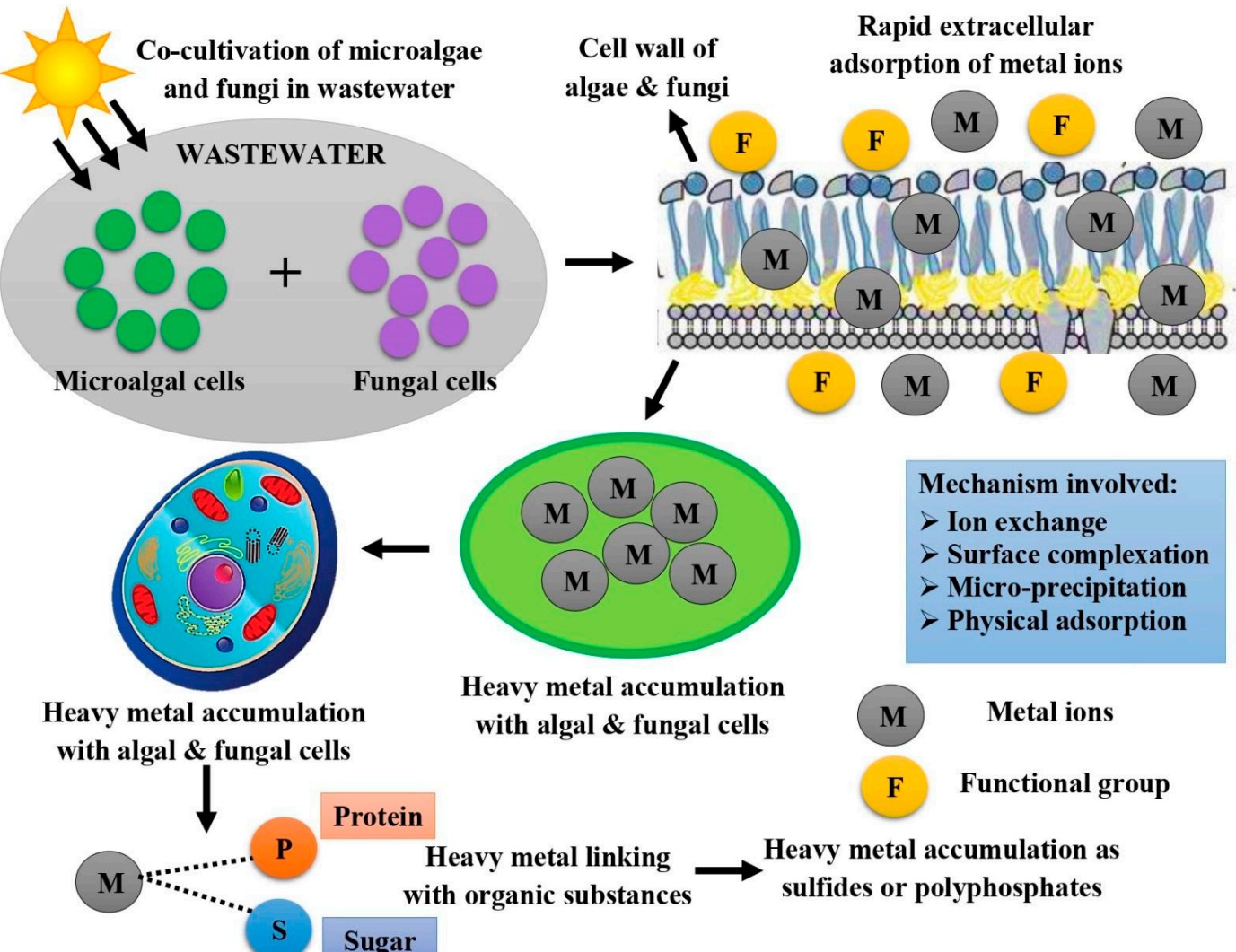

**Figure 6.** Removal of heavy metal ions from wastewater by utilizing a microalgae–fungi consortium.

## 5. Utilization of Algae and Its Consortium for Wastewater Treatment

The high cost requirement and energy consumption of conventional systems for wastewater treatment has necessitated the utilization of economical, environmentally friendly and sustainable wastewater treatment systems, for which the use of microalgae can be seen as a promising approach, because of its efficiency, cost-effectiveness and economical nature [53,99,100]. Microalgae significantly remove various organic pollutants, mainly phosphorus and nitrogen, followed by their useful conversion into compounds such as lipids and proteins [14,101]. Apart from nutrient removal, microalgae are also responsible for the bio-absorption of various hazardous heavy metals, including arsenic, chromium, lead, mercury and cadmium [101]: for example, a microalgae–fungi consortium involving *Chlorella vulgaris* and *Aspergillus oryzae* significantly removed arsenic by 51.14% in wastewater [102]; similarly, Cr (III) was effectively removed by 99% through an algae–algae (*Tetradesmus* sp., *Scenedesmus* sp. and *Ascomycota* sp.) consortium [54]. Recently, pure microalgae strains have been explored for treating multiple wastewaters; however, the association of algae and other microorganisms is regarded as a more promising approach

than the monosystem [99]. Moreover, these consortiums can overcome issues related to wastewater treatment, such as irregular removal efficiency, increased cost of treatment, and low biodegradability of some pollutants, and can therefore be a positive option for phycoremediation, as discussed below [103].

### 5.1. Utilization of Algae–Algae Consortiums for Wastewater Treatment

More recently, scientific studies have further outlined the capability of microalgae consortiums (involving only algal species) in distinct applications, involving nutrient removal and biomass production. The utilization of microalgae consortiums in wastewater treatment processes can be a robust system that is able to withstand varying environmental conditions and interference by other species. Moreover, the system facilitates broad specificity to multiple nutrients, i.e., the association of microorganisms with various nutrient demands simultaneously leads to the remediation of nutrients. In addition, the cooperative associations further result in enhanced removal efficacy, and can be utilized in the tertiary treatment step of wastewater treatment, thus promoting the efficient removal of phosphorus, nitrogen and other contaminants like heavy metals [15]. Many studies have analyzed the possible potential of microalgae consortiums in wastewater treatment; for example, Koreivienė et al. [104] developed a consortium of *Scenedesmus* sp. and *Chlorella* sp. for the removal of inorganic phosphorus (IP) and inorganic nitrogen (IN) from municipal wastewater, which was 1.04 to 4.17 mg/L and 56.5 mg N/L initially, but significantly, after treatment, was removed by >99% and 88.6–96.4% Similarly, Chinnasamy et al. [105] observed the complete remediation of $PO_4$-P and $NO_3$-N, with removal efficiency of 99.8% and 96.6% in carpet mill effluents; moreover, high lipid and biomass productivity of about 6.82% and 9.2–17.8 ton ha$^{-1}$ yr$^{-1}$ was also recorded. Likewise, Renuka et al. [9] estimated the efficiency of filamentous and unicellular microalgae consortium for treating primary-treated sewage, and revealed a high removal rate for phosphorus and nitrogen: in particular, the removal efficacy of $NO_3$-N, $NH_4$-N and $PO_4$-P ranged between 81.5 to 83.3%, 100% and 94.9 to 97.8%, respectively. Utilization of algae–algae consortium for wastewater treatment is detailed in Table 1.

**Table 1.** Utilization of algae–algae consortium for wastewater treatment.

| S.No. | Algal Species Used | Pre-Culture | Cultivation Conditions | Type of Wastewater | Source of Wastewater | Target Pollutant/Physicochemical Characteristics | Removal Efficiency (%) | Reference |
|---|---|---|---|---|---|---|---|---|
| 1. | *Chlorella sorokiniana, Chlorella vulgaris, Scenedesmus obliquus* | 0.04 g drybiomss/L | In WPW $27 \pm 2$ °C with a photoperiod of 16 h light: 8 h dark | Meat processing wastewater | Beef packaging plant in Nebraska, USA | COD<br>TN<br>TP-$PO_4^{3-}$ | 91<br>67<br>69 | [53] |
| 2. | *Chlorella saccharophila, Chlamydomonas pseudococcum, Scenedesmus* sp., *Neochloris oleoabundans* | 0.1 g $L^{-1}$ | In BG-11 at light intensity of 80 mmol $m^{-2}$ $s^{-1}$ at 30 °C with 12 h light: 12 h dark for two weeks | Dairy wastewater | Dairy farm in Perlis, Malaysia | BOD<br>COD<br>TSS<br>TDS<br>TKN<br>$NH_4$-N<br>$NO_3$-N<br>$PO_4$-P | 82.60 to 83.14<br>88.90 to 89.02<br>86.25 to 76.16<br>77.23 to 80.40<br>98.33 to 97.83<br>99.61 to 98.00<br>96.97 to 89.93<br>93.02 to 88.84 | [55] |
| 3. | *Tetraselmis* sp. | - | 24 h of illumination under constant aeration with a flow of 1 L $min^{-1}$ | Tannery wastewater | Leather finishing stage in Novo Hamburgo, Brazil | TN<br>P-$PO_4$<br>TOC<br>COD<br>BOD<br>$NH_3$ | 71.74<br>97.64<br>31.35<br>56.70<br>20.68<br>100 | [106] |
| 4. | *Coelastrum microporum* | 4.0 g/L | Light intensity of 120 μmol $m^{-2}$ $s^{-1}$ with 12 h light: 12 h dark at 25 °C in PBR with aeration at 0.2 vvm by globular stone. | Activated sludge (primary influent to the WWTP) | Daejeon Metropolitan City Facilities Management Corporation in Daejeon, Korea | TDN<br>TDP<br>SCOD | 97.0<br>98.3<br>77.1 | [56] |
| 5. | *Tetradesmus* sp., *Scenedesmus* sp. and *Ascomycota* sp. | - | In BBM at room temperature, pH 8 with light intensity of 20 μmol $m^{-2}$ $s^{-1}$ in PBR | Tannery wastewater | Tannery industry in Mexico | Cr (III) | 99 | [54] |
| 6. | *Scenedesmus* sp., other species of green algae, Cyanobacteria, diatoms | Mixed Culture ($93 \pm 2\%$; $4 \pm 1\%$; $2 \pm 1\%$; $1 \pm 0.01\%$) respectively. | Light intensity of 220 μmol $m^{-2}$ $s^{-1}$ at $27 \pm 2$ °C with a photoperiod of 12 h light: 12 h dark in PBR at Ph 8.5 | Digestate and secondary effluent | Lab-scale microalgae anaerobic digester and secondary settler treating urban wastewater | TN<br>TP<br>TOC | 58<br>83<br>85 | [107] |
| 7. | *Chlorella, Scenedesmus, Chaetophora* and *Navicula* | - | The microalgae were grown in PBR under natural light and temperature | Urban wastewater | Águas da Figueira (AdF, Figueira da Foz, PT) | $NH_4^+$<br>P<br>COD | 99.5<br>100<br>40.64 | [108] |
| 8. | *Chlorella vulgaris, Chlorella protothecoides* | 40 g/L | In PBR with 1000 L of de-chlorinated tap water and 10 g of synthetic fertilizer at pH 8.8 under 27 to 28 °C | Municipal Wastewater | WWTP in South Africa | TN<br>TP<br>TOC<br>COD<br>Orthophosphate | 35.4<br>74.4<br>22.2<br>60.0<br>87.0 | [109] |

**Table 1.** *Cont.*

| S.No. | Algal Species Used | Pre-Culture | Cultivation Conditions | Type of Wastewater | Source of Wastewater | Target Pollutant/Physicochemical Characteristics | Removal Efficiency (%) | Reference |
|-------|-------------------|-------------|------------------------|--------------------|--------------------|--------------------------------------------------|------------------------|-----------|
| 9. | *Chlorella* sp., *Merismopedia* sp., *Closteriopsis* sp., *Scenedesmus* sp. | 10 mL/100 mL | In BG-11 medium at $25 \pm 2$ °C with a light intensity of 4.5 Klux | Gray water | Drainage line at IIT Delhi, India | TDP<br>TAN<br>$NO_3$-N<br>COD | 98.28<br>88.23<br>86.55<br>82.45 | [110] |
| 10. | *Chlorella vulgaris, Chlorella protothecoides* | 40 g/L | In PBR with 1000 L of de-chlorinated tap water and 10 g of synthetic fertilizer at pH 9.1 under 29 to 31 °C | Municipal Wastewater | WWTP in South Africa | TN<br>TP<br>TOC<br>COD<br>Orthophosphate | 73.1<br>50.0<br>54.0<br>6.6<br>83.0 | [109] |
| 11. | *Scenedesmus quadricauda, Euglena gracilis, Chlorella vulgaris, Ankistrodesmus convolutes, Chlorococcum oviforme* | 10% from exponential phase | In BBM constituting 0, 25, 50, 75 and 100% of leachate under 42 µmol photons m$^{-2}$ s$^{-1}$ of irradiance with a photoperiod of 12 h light: 12 h dark at $25 \pm 1$ °C | Landfill leachate | Sanitary landfill in Selangor, Malaysia | $NH_4$-N<br>COD<br>$PO_4$-P | 92.01 to 98.73<br>69.41 to 90.97<br>44.93 to 85.97 | [111] |
| 12. | *Chlorella* sp., *Scenedesmus* sp., *Sphaerocystis* sp., *Spirulina* sp. | $10^5$ cells mL$^{-1}$ equivalent to DBW 0.13 g/L | In 90 mL of CHU 10 medium inoculated with 10 mL of wastewater at $31 \pm 1$ °C under 16 h light: 8 h dark with 80 mmol m$^{-2}$ s$^{-1}$ and 50% (*v/v*) of $CO_2$ | Domestic wastewater | Facultative pond at domestic WWTP in Kalyani, India | $CO_2$<br>$PO_4$-P<br>$NH_4$-N | 53 to 100<br>59<br>39 | [112] |
| 13. | *Phormidium* and *Chlorella pyrenoidosa* | 300 mL | Initially maintained in ACA and then transferred to slants in BG 11 medium under $70 \pm 5$ µmol m$^{-2}$ s$^{-1}$ at $25 \pm 2$ °C | Municipal wastewater | Drain in IIT Delhi | COD<br>TAN<br>TDP<br>$NO_3$-N | $53 \pm 2$<br>$81 \pm 3$<br>$75 \pm 2$<br>$87 \pm 5$ | [113] |

COD—chemical oxygen demand; TN—total nitrogen; TP-$PO_4^{3-}$—total phosphate; WPW—whole processing wastewater; P-$PO_4$—orthophosphate; TOC—total organic carbon; BOD—biological oxygen demand; $NH_3$—ammonia; BBM—Bold's Basal Medium; NMC—native microalgae consortium; Cr (III)—chromium (III); TSS—total suspended solids; TDS—total dissolved solids; TKN—Total Kjeldahl Nitrogen; $NH_4^+$-N—nitrogen content of ammonium ion; $NO_3$-N—nitrate nitrogen; P—phosphorus; $NH_4^+$—ammonium; PBR—photobioreactor; TDN—total dissolved nitrogen; TDP—total dissolved phosphorus; SCOD—soluble chemical oxygen demand; WWTP—wastewater treatment plant; TAN—total ammonia nitrogen; IIT—Indian Institute of Technology; ACA—algae culture agar; DBW—dry biomass weight.

### 5.2. Utilization of Algae–Bacterial Consortiums for Wastewater Treatment

In the process of wastewater treatment, a microalgae–bacterial consortium offers certain advantages, such as minimizing the emission of greenhouse gasses, the effective removal of pollutants, and cost-efficient aeration. Moreover, algae can also eliminate pathogens, including viruses, and the generation of bacteria and algae flocs at the time of wastewater treatment, further assisting the easy downstream processing of biomass through sedimentation, and thus excluding the use of flocculating agents [88,114,115]. Existing studies have conveyed that the advanced and enhanced removal of nutrients from wastewater can be attained by the symbiosis of algae and bacteria, in contrast to the monoculture of bacteria or algae. In addition, this system can further increase the recovery of biofertilizer from wastewater treatment plants [116,117]. During treatment, the microalgae–bacteria consortium remediates heavy metals or other organic pollutants by mechanisms such as bioaccumulation, biosorption, and biodegradation [23]. The symbiotic association between algae and bacteria has been utilized in treating municipal wastewater, saline wastewater, domestic wastewater, pharmaceutical wastewater, wastewater contaminated with heavy metals, chemical industry wastewater, piggery wastewater, aquaculture wastewater and many more [23,77,118,119].

For instance, Biswas et al. [120] revealed that microalgae–bacterial consortiums possessed significant potential for dairy wastewater remediation along with high lipid biomass productivity: their study observed a significant reduction in chemical oxygen demand (COD), ammonium, and nitrates and phosphates by 93%, 87.2% and 100%, respectively, after 48 h of treatment at $25 \pm 2\ ^\circ C$; in addition, the biomass productivity was enhanced by 67%, exhibiting 42% of lipid, 55% of carbohydrates and 18.6% of protein content. Similarly, Da Silva Rodrigues et al. [121] observed that Sulfamethoxazole (SMX) was effectively removed by $54.34 \pm 2.35\%$ from wastewater treatment plant effluents through a microalgae–bacteria consortium: this removal process may have been associated with symbiotic biodegradation by bacteria, owing to the rise of oxygen released by the photosynthetic process of the microalgae; thus, the study demonstrated a promising substitute for bioremediation of SMX. Yang et al. [122] applied an algal–bacterial consortium in a photo membrane bioreactor for wastewater treatment, and noticed that ammonium and COD were significantly removed by 100% and 90%; in addition, the phosphate removal was around 3 mg $PO_4^{3-}$-P/L.h. Foladori et al. [99] also utilized a microalgae–bacterial consortium for treating real municipal wastewater, and observed a significant removal of $86 \pm 2\%$ and $97 \pm 3\%$ in COD and TKN of treated wastewater. Likewise, Posadas et al. [123] utilized microalgae–bacterial consortiums for treating wastewaters from five different agro-industries: potato processing wastewater (PW); fish processing wastewater (FW); industry producing animal food (MW); lyophilized coffee manufacturing wastewater (CW); and wastewater from a yeast production factory previously subjected to anaerobic digestion (YW); they observed the maximum removal of total organic carbon ($64 \pm 2\%$) and nitrogen ($85 \pm 1\%$) in 2-fold diluted FW, while P-$PO_4^{3-}$ was removed by $89 \pm 1$ % in undiluted PW. Utilization of microalgae–bacterial consortiums for wastewater treatment is detailed in Table 2.

**Table 2.** Utilization of microalgae–bacterial consortium for wastewater treatment.

| S.No. | Algal Strain Used | Bacterial Strain Used | Way of Cultivation | Reactor Type | Cultivation Conditions | Type of Wastewater | Source of Wastewater | Target Pollutant/Physicochemical Characteristics | Removal Efficiency (%) | Reference |
|---|---|---|---|---|---|---|---|---|---|---|
| 1. | *Scenedesmus obliquus* and *Chlorella vulgaris* | *Raoultella terrigena* and *P. agglomerans* | Batch | Pilot-scale PBR | 14.5 L of synthetic medium for OWW+ 1.5 L of consortium with $10^{12}$ cells mL$^{-1}$ and $10^3$ CFU mL$^{-1}$ of microalgae and bacteria at $25 \pm 1$ °C, 160 rpm rotation with light intensity of 200 µmol m$^{-2}$ s$^{-2}$ for 16:8 h light-dark cycle for 48–72 h | Olive-washing water | Olive oil factory of Spain | TPC COD BOD$_5$ Turbidity Color | $90.3 \pm 11.4$ $80.7 \pm 9.7$ $97.8 \pm 12.7$ $82.9 \pm 8.4$ $83.3 \pm 10.4$ | [124] |
| 2. | *Tetraselmis indica* | *Pseudomonas aeruginosa* | Batch | 500 mL Erlenmeyer flasks | Light intensity of 130 µmol/(m$^2$s) with a 16 h/8 h light/dark cycle at 28 °C for 10 days | Dairy wastewater | Kwality Ltd., dairy processing plant in Saharanpur, India | COD TDN TDP | 87.49 83.76 79.83 | [64] |
| 3. | *Microcystis* sp., *Oscillatoria* sp., *Chlorella* sp., *Scenesdesmus* sp., *Stigeoclonium* sp. | Strain was not specified | Batch | 10 L- PBR | Continuous illumination of 76 µmol m$^{-2}$ s$^{-1}$ and 5 L loading with 10% (*v/v*) diluted landfill leachate at $25 \pm 1$ °C, 5.0–8.0 mg/L of DO with 6.5–8.5 of pH | Landfill leachate | Northern Cyprus leachate storage tank | TN P-PO$_4^{3-}$ COD Phenol | 99.4 98.88% to 99.39 90.1 to 92.34 99.55 | [125] |
| 4. | *Chlorella pyrenoidosa* | Strain was not specified | Batch | 500 mL flasks | 400 mL of municipal wastewater spiked with 0%, 5%, 10%, 15%, 20% of leachate inoculated with 0.05 g L$^{-1}$, at 25 °C, light intensity of 8000 Lux | Municipal wastewater and landfill leachate | Grit chamber at Quyang Wastewater Plant and Laogang Landfill in Shanghai, China. | NH$_4^+$-N P | 95 <95 | [126] |

**Table 2.** *Cont.*

| S.No. | Algal Strain Used | Bacterial Strain Used | Way of Cultivation | Reactor Type | Cultivation Conditions | Type of Wastewater | Source of Wastewater | Target Pollutant/Physicochemical Characteristics | Removal Efficiency (%) | Reference |
|---|---|---|---|---|---|---|---|---|---|---|
| 5. | *Scenedesmus obliquus* | *Bacillus megaterium* | Batch | 500 mL conical flask | Microalgae and bacteria at a concentration of $3 \times 10^5$ cells/mL and $1 \times 10^5$ cells/mL in 200 mL of biogas slurry at $25 \pm 2$ °C with light intensity of $45$ $\mu$mol/m$^2$/s and light:dark cycle of 14 h:10 h | Biogas slurry | Anaerobic tank of a pig farm in Yantai, Shandong province | COD TP NH$_4^+$-N | 85.98 81.03 65.48 | [65] |
| 6. | *Chlorella* sp., *Chlamydomonas* sp. and *Scenedesmus* sp. | Strain was not specified | Batch | 1 L of bioreactor | Microalgae–bacteria consortium was prepared at a fixed ratio of 18% culture to wastewater by volume with a light intensity of $120$ $\mu$E/m$^2$s at room temperature | Municipal wastewater | WWTP in Akaki Kality sub city of Addis Ababa, Ethiopia | TKN TP PO$_4^{3-}$-P COD BOD$_5$ | 69 59 73 84 85 | [127] |
| 7. | *Scenedesmus* sp. | Strain was not specified | Batch | PBR | Microalgae and activated sludge were mixed in the ratio of 1:0; 0:1; 1:1; 1:3; 1:5 and 3:1 with a constant air injection of 2 L/min at $25 \pm 2$ °C with 12 h:12 h of light-dark cycle at $200$ $\mu$mol/m$^2 \cdot$s at a pH of $7.5 \pm 0.5$ | Paper pulp Wastewater | Paper pulp industry WWTP in Portugal | COD PO$_4^{3-}$-P NH$_4^+$-N | 85.50 86 86.81 | [128] |

**Table 2.** *Cont.*

| S.No. | Algal Strain Used | Bacterial Strain Used | Way of Cultivation | Reactor Type | Cultivation Conditions | Type of Wastewater | Source of Wastewater | Target Pollutant/Physicochemical Characteristics | Removal Efficiency (%) | Reference |
|---|---|---|---|---|---|---|---|---|---|---|
| 8. | *Chlorella vulgaris* and *Scenedesmus obliquus* | *Proteobacteria, Firmicutes, Bacteroidetes* and *Chloroflexi* | Batch | PBR | Algae:sludge inoculation ratio was 1:1 (*w/w*) with a light intensity of 40 to 50 $\mu$mol.m$^{-2}$ s$^{-1}$ at 100 rpm with no pH control and aeration at a flow rate of 15 L h$^{-1}$. Temperature and photoperiod were 31.2 °C (light): 20.5 °C (dark) of a 14.2 h: 9.8 h light/dark cycle and 25.8 °C (light): 16.9 °C (dark) of a 12.4 h: 11.6 h light/dark cycle | Municipal wastewater | Aerated grit chamber in third sewage treatment plant of China | COD NH$_4^+$ PO$_4^{3-}$ TSS | 93.7 $\pm$ 0.9 100.0 $\pm$ 0.0 98.4 $\pm$ 1.5 96.3 $\pm$ 2.1 | [129] |
| 9. | *Chlorella vulgaris* | *Exiguobacterium* and *B. licheniformis* | Batch | 1.0 L columnar PBR | Algae:bacteria inoculation ratio were 1:0:0; 1:2:0; 1:0:2; 1:1:1 and the amounts of *Chlorella* and bacteria were 6.8 $\times$ 10$^6$ cells mL$^{-1}$ and 13.6 $\times$ 10$^6$ CFU mL$^{-1}$ with a light intensity of 120.0 $\mu$mol photons m$^{-2}$ s$^{-1}$ at temperature 25.0 $\pm$ 1.0 °C with 0.3 L m$^{-1}$ of ventilation rate | Piggery wastewater | Yantai Longda Breeding Co., Ltd., in Yantai, Shandong | TN TP NH$_4^+$-N COD | 78.3 87.2 84.4 86.3 | [67] |

**Table 2.** *Cont.*

| S.No. | Algal Strain Used | Bacterial Strain Used | Way of Cultivation | Reactor Type | Cultivation Conditions | Type of Wastewater | Source of Wastewater | Target Pollutant/Physicochemical Characteristics | Removal Efficiency (%) | Reference |
|---|---|---|---|---|---|---|---|---|---|---|
| 10. | *Chlamydomonas reinhardtii*, *Chlorella vulgari* and *Scenedesmus obliquus* | Strain was not specified | Batch | 2 L glass bottles with 3 ports lids (air inlet and outlet ports topped with 0.45 µm filter to avoid contamination and sampling port) | Algal concentration was 0.20–0.25 g/L in 1.8 L of brewery wastewater at 20 °C with light intensity 70 µmol photons $m^{-2}s^{-1}$ with 12 h light/12 h dark and 100 rpm consistent mixing | Brewery wastewater | — | COD<br>TN<br>TP | >85<br>>80<br>>70 | [130] |
| 11. | *Chlorellaceae* sp., *Scenedesmaceae* sp., *Chlamydomonadaceae* sp. | Strain was not specified | Batch | Outdoor high-rate algal pond (or raceway pond) | Microalgal species at a concentration of $1.10^6$ cells.$mL^{-1}$; $0.2 \times 10^6$ cells.$mL^{-1}$ and $0.2 \times 10^6$ cells $mL^{-1}$ in 500 L of piggery wastewater and 340 L of tap water | Piggery wastewater | Piggery farm in Cremona Province (Po Valley, Northern Italy) | NH$_4^+$-N<br>Orthophosphate<br>COD | 90<br>90<br>59 | [131] |
| 12. | *Chlorella* sp. | *Acinetobacter* sp. | Batch | Pilot-scale bioreactor | Algal cells with a density of $0.275 \pm 0.025$ g/L were inoculated in 100 mL of centrate wastewater at a light intensity $120 \pm 10$ µmol photons $m^{-2}s^{-1}$ at $25 \pm 1$ °C with relative humidity $45 \pm 3\%$ at 200 rpm | Centrate wastewater | Municipal WWTP in St. Paul (Minnesota, USA) | COD<br><br>TP | 93.01<br><br>98.78 | [132] |

**Table 2.** *Cont.*

| S.No. | Algal Strain Used | Bacterial Strain Used | Way of Cultivation | Reactor Type | Cultivation Conditions | Type of Wastewater | Source of Wastewater | Target Pollutant/Physicochemical Characteristics | Removal Efficiency (%) | Reference |
|---|---|---|---|---|---|---|---|---|---|---|
| 13. | *Chlorella* sp. | *Bacillus firmus* and *Beijerinckia fluminensis* | Batch | 500 mL Erlenmeyer flasks | Concentration of algae and bacteria were $1.0 \times 10^5$ cells/mL and 1% (*v/v*) or 10% (*v/v*) at 26 °C with light intensity of $50 \pm 10$ μmol/(m$^2$s) in 200 mL of vinegar fermentation wastewater | Vinegar production wastewater | Hengshun Vinegar Industry Co., Ltd., Zhenjiang, Jiangsu, China | COD TN TP | 22.1 20.0 18.1 | [133] |

COD—chemical oxygen demand; TDN—total dissolved nitrogen; TDP—total dissolved phosphorus; P-PO$_4^{3-}$—orthophosphate; TN—total nitrogen; TP—total phosphorus; NH$_4^+$-N—nitrogen content of ammonium ion; BOD—biological oxygen demand; TKN—Total Kjeldahl Nitrogen; WWTP—wastewater treatment plant; N—nitrogen; P—phosphorus; TPC—total phenol concentration; TSS—total soluble solids; OWW—olive-washing wastewater; CFU—colony-forming unit; DO—dissolved oxygen.

### 5.3. Utilization of Algae–Fungi Consortiums for Wastewater Treatment

In recent years, the potentiality of great tolerance and increased agglomeration in microalgae–fungi consortiums has contributed to enlightenment in the treatment of wastewater contaminated with heavy metals or other pollutants. Apart from being a successful method for removing various pollutants from wastewater, the co-cultivation of microalgae and fungi further assists the easy harvesting of microalgae [22]. In particular, microalgae–fungi consortiums possess the efficiency to treat wastewater contaminated with pharmaceuticals and dyes. Microalgae are extensively utilized to treat antibiotics by photodegradation, adsorption, biodegradation, hydrolysis and accumulation [134]. Moreover, fungi that have adsorption characteristics, and produce extracellular and intracellular enzymes, can successfully treat pesticides, phenols, antibiotics, dyes and similar organic micropollutants in wastewater [26]. Co-cultivation technology therefore has a double purpose, in decontaminating wastewater discharged from various sources, and in accumulating microalgae-derived biomass products, thus forming a circular bio-economy. Many existing scientific studies have researched the potential of microalgae–fungi consortiums for decontaminating wastewater. For instance, Wrede et al. [94] utilized microalgae–fungi consortiums for treatment of anaerobically digested swine lagoon wastewater, and observed the potential for subsequent wastewater purification, thus improving the economics of mass-scale algal biotechnology; in addition, the yield of total lipid content was also improved. Similarly, Zhang et al. [135] also implemented microalgae–fungi consortium under mixed LED light wavelengths, by utilizing the species *Chlorella vulgaris* and *Ganoderma lucidum* for purification of biogas slurry received from an anaerobic digestion reactor of Jiaxing pig farm, Zhejiang (China): their study observed that under a ratio of red: blue light, the COD, TN, and TP were significantly eliminated by $76.35 \pm 6.87\%$, $78.77 \pm 7.13\%$ and $79.49 \pm 7.43\%$, respectively. Likewise, Wang et al. [136] treated starch wastewater with a microalgae-fungi consortium, and observed that the removal efficiencies of TP, TN, and COD reached 92.08, 83.56, and 96.58 %, respectively. Utilization of microalgae–fungi consortiums for wastewater treatment is detailed in Table 3.

**Table 3.** Utilization of microalgae–fungi consortium for wastewater treatment.

| S.No. | Algal Strain Used | Fungal Strain Used | Type of Wastewater | Source of Wastewater | Target Pollutant/Physicochemical Characteristics | Removal Efficiency (%) | Reference |
|---|---|---|---|---|---|---|---|
| 1. | *Chlorella pyrenoidosa* | *Rhodosporidium toruloides* | Rice wine distillery wastewater and domestic wastewater | S1 distillery in Foshan, China and local wastewater treatment plant in Macau, China | SCOD<br>TN<br>TP | $95.34 \pm 0.07$<br>$51.18 \pm 2.17$<br>$89.29 \pm 4.91$ | [82] |
| 2. | *Scenedesmus obliquus* | *Trichoderma reesei* | Municipal wastewater | Effluent of a treatment plant in Mexico | Nitrate<br>TAN<br>Orthophosphate | 96<br>100<br>93 | [137] |
| 3. | *Chlorella vulgaris* | *Aspergillus* sp. | Swine manure wastewater | Umore Park (Rosemount, MN, USA) | Ammonium<br>TN<br>TP<br>COD | 23.23<br>44.68<br>84.70<br>70.34 | [83] |
| 4. | *Chlorella vulgaris* | *Ganoderma lucidum* | Biogas slurry | Anaerobic digestion reactor in a livestock WWTP in Jiaxing pig farm, Zhejiang, China | COD<br>TN<br>TP<br>$CO_2$ | $92.17 \pm 5.28$<br>$89.83 \pm 4.36$<br>$90.31 \pm 4.69$<br>$74.26 \pm 3.14$ | [93] |
| 5. | *Chlorella vulgaris* | *Aspergillus Niger* | Artificially prepared wastewater | — | Ranitidine | $50 \pm 19$ | [138] |
| 6. | *Chlorella vulgaris* | *Aspergillus oryzae* | Artificially prepared wastewater | — | Arsenic | 51.14 | [102] |
| 7. | *Scenedesmus* sp. | *Trichoderma reesei* | Secondary effluent | Seafood processing plant | COD<br>TN<br>TP | >74<br>>44<br>>93 | [78] |

**Table 3.** *Cont.*

| S.No. | Algal Strain Used | Fungal Strain Used | Type of Wastewater | Source of Wastewater | Target Pollutant/Physicochemical Characteristics | Removal Efficiency (%) | Reference |
|---|---|---|---|---|---|---|---|
| 8. | *Tetradesmus obliquus* | *Aspergillus niger* | Gold mining wastewater | Tailing of Sibanye Stillwater in South Africa | Gold | 97.77 | [139] |
| 9. | *Chlorella vulgaris* | *Aspergillus* sp. | Molasses wastewater | Local plant in Guangzhou, China | Color<br>COD<br>TP<br>TN<br>$NH_3$-N | 69.98<br>70.68<br>88.39<br>67.09<br>94.72 | [140] |
| 10. | *Chlorella vulgaris* | *Ganoderma lucidum* | Biogas slurry | — | COD<br>TN<br>TP | 70<br>75<br>78 | [141] |
| 11. | *Chlorella vulgaris* | *Ganoderma lucidum* | Biogas slurry | Anaerobic digester in Hongmao Hacienda, Kunshan City, China | COD<br>TN<br>TP<br>$CO_2$ | 68.29<br>61.75<br>64.21<br>64.68 | [141] |
| 12. | *Chlorella vulgaris* | *Ganoderma lucidum* | Anaerobically digested swine wastewater | Anaerobic digestion reactor in a livestock WWTP of pig farm in Jiaxing, Zhejiang, China | COD<br>TN<br>TP | $79.74 \pm 4.87$<br>$74.28 \pm 6.13$<br>$85.37 \pm 6.84$ | [142] |
| 13. | *Chlorella sorokiniana* | *Aspergillus niger* | Municipal wastewater | Prem Nagar sewer system, Dehradun, Uttarakhand, India | TKN<br>BOD<br>COD<br>TOC | 95.40<br>81.78<br>83.67<br>70.26 | [143] |
| 14. | *Scenedesmus abundans* | *Saccharomyces Cerevisiae* | Dairy wastewater | Graphic Era University dairy, Uttarakhand, India | TN<br>TP<br>COD<br>BOD | 41.7<br>60.9<br>83<br>90 | [144] |

SCOD—soluble chemical oxygen demand; TN—total nitrogen; TP—total phosphorus; TAN—total ammonia nitrogen; COD—chemical oxygen demand; $CO_2$—carbon dioxide; WWTP—wastewater treatment plant; $NH_3$-N—nitrogen content of ammonia; TKN—Total Kjeldahl Nitrogen; TOC—total organic carbon.

## 6. Flocculation of Algal Consortiums

Apart from removing pollutants from wastewater, microalgal-based wastewater treatment systems further contribute to the production of valuable microalgal biomass that can be valorized for different purposes, such as biofertilizers [145]. In recent years, flocculation has been seen as one of the most practicable techniques for harvesting algal biomass on a commercial scale. Flocculation is a method in which the cells dispersed in aqueous culture come closer to each other to produce large aggregates with enhanced settling velocity, thus resulting in easy harvesting of algal biomass through gravity sedimentation [146]. Sometimes, microalgae present in water reserves, such as rivers, ponds and lakes, undergo the process of flocculation on their own, due to the extracellular polymeric compounds (EPS) in the medium, generated by other microorganisms including fungi or bacteria: this is known as bio-flocculation [147]. The fungal species can interact with the negatively charged surface of microalgae by their positively charged hyphae, to induce flocculation; similarly, bacteria can also lead to flocculation. A consortium of algae with bacteria or fungi demands a carbon source, which can be naturally found in wastewater: therefore, this consortium system can be used to harvest microalgae at the time of wastewater treatment [146]; however, there are no data on the settling characteristics of flocculated microalgae [148]. Very few studies have explored certain microalgae physical properties, such as concentration factor, floc size and settling velocity [148,149]. The distribution of the settling velocity of flocculated algal biomass is an essential parameter for designing cost-efficient gravity settlers for recovery of biomass. In high-rate algal ponds, critical settling velocity reduces steadily in successive columns, because of the gradual rise in column diameter, thus leading to the retention of biomass flocs in various columns on the basis of their settling velocity; therefore, flocs

which have a greater or equal settling velocity than the critical settling velocity of a given column will retain, whereas the flocs with low settling velocity will escape to the following column [145]. The critical settling velocity can be calculated by

$$Vi = Q/Si$$

where Vi = critical settling velocity (m/h), Q = flow rate ($m^3$/h) and Si = area of column ($m^2$).

## 7. Factors Affecting Wastewater Treatment by Algal System

For the actual application of the algal system to the wastewater treatment, lighting during night-time, mixing, the depth of the algal tank, and the recycle ratio of the settled algal sludge are some of the important parameters, as discussed below.

### 7.1. Lighting at Night-Time

Light is a crucial factor in microalgae cultivation [150]: photoperiods, light frequency and light intensity have been reported to affect the efficiency of nutrient removal and microalgae productivity [151,152]. In general, the growth rate of microalgae is proportional to intensity before the saturation point is reached, at which point the photosynthetic mechanism of the microalgae attains its highest value [153]; however, when it is reduced below its optimal value, the growth of the microalgae is restricted [154,155]. On the other hand, when the light intensity surpasses its optimum value, photosystem I and photosystem II can be damaged, leading to photo-inhibition in the microalgae [152,156]: this photo-inhibition can be minimized by uniting periods of high light intensity with periods of darkness [153]. The short-term absence of light is thought to permit the photosynthesis of dark reactions that are slower than light reactions for utilizing the energy stored from dark reactions. In reality, the excess photons absorbed by the microalgae are released as fluorescence or heat, and decrease the efficiency of the photosynthesis; therefore, the utilization of appropriate light:dark photoperiods has been described to decrease the demand of light energy by the same, or sometimes to increase with similar or even higher productivity [157]. For example, Habibi et al. [158] studied the effect of different light/dark cycles (i.e., 12/12, 16/8 and 24/0 h) for phosphate and nitrate removal from artificially prepared wastewater, by utilizing Scenedesmus sp.: their study observed the maximum removal of nitrate and phosphate in slaughterhouse and dairy synthetic wastewater by 78% and 99.7% and by 31% and 68% after 24/0 h of the photoperiod.

### 7.2. Mixing

Mixing is also an important parameter that influences the growth of microalgae culture, as it permits the equal distribution of nutrients and light between the algal cells—hence ignoring the existence of stagnant areas—and enhances the rate of gas transfer between the air and the culture medium. The rate of gas transfer should not be compromised, because the air bubbled into the microalgae cultures constitutes carbon dioxide, which is necessary for photosynthesis, and which eliminates the generated oxygen. Moreover, mixing is also essential for preventing the settling of microalgae cultures and thermal stratification [15]. The process of mixing involves the movement of algae from high illuminated areas of the reactor to dark zones, thus minimizing photo-inhibition [157].

### 7.3. Depth of Algal Tank

Wastewater treatment systems include large shallow ponds, circular ponds and tanks, but the most common system utilized for wastewater treatment is the raceway pond [159]. The working depth of an algal tank is an essential design parameter of raceway ponds: this is because designing the tank with a shallow depth can expose algae to higher temperatures, mainly during summer. On the other hand, a very high pond depth can hinder the sufficient penetration of light. The ideal depth of the algal tank can be decided on the basis of the quantity and quality of the light, and the turbidity of the wastewater to be treated, which promotes light-scattering attenuation and processes [160]. In general, the depth of high-

rate algal ponds varies between 10 to 50 cm. For example, Kim et al. [161] studied the effect of water depth (20, 30, 40 cm) on the nutrient removal efficiency of *Stigeoclonium* sp., *Scenedesmus* sp., and *Chlorella* sp. from municipal wastewater: their study observed that the removal efficiency of total nitrogen and total phosphorus was 82.5%, 43.4%, and 18.6% and 89.7%, 36.0%, and 32.3%, respectively, for 20, 30, and 40 cm depth tanks; hence, in a 20 cm depth tank the removal efficiency was maximum.

*7.4. Recycle Ratio of Settled Algal Sludge*

Microalgal species in high-rate algal ponds settle naturally, due to gravity, as soon as they are removed from the mixing of the algal ponds, into shorter hydraulic retention time (HRT) algal harvest tanks or simple algal settling ponds. Such ponds allow the natural settling of algal biomass, and further contribute to the storage of settled algae for periodic recovery. The removal efficiency of microalgae can be enhanced by their aggregation/flocculation when carbon dioxide is added to the high-rate algal ponds or when a proportion of settled algae is recycled back to the high-rate algal ponds in the same manner as sludge is recycled in the activated sludge process [162].

## 8. Observed Yield Coefficient

The observed yield coefficient (Yobs) in sludge processing plants can be stated as a measure of the biomass that is the mixed liquid and suspended solids, synthesized by a given biological oxygen demand (BOD) [163]: in other words,

$$\text{Yobs} = \Delta\text{MLSS}/\Delta\text{BOD} \tag{1}$$

where Yobs = the observed yield coefficient, $\Delta$MLSS = mixed liquid and suspended solids, and $\Delta$BOD = biological oxygen demand.

The observed yield coefficient is an essential parameter in mathematical models utilized in wastewater treatment systems, such as Aerobic Activated Sludge Model 3 and Aerobic Activated Sludge Model 1; it can also be used for estimating the kinetic parameters, such as the highest specific growth rate in the treatment system. Therefore, a process that could effectively find out the observed yield coefficient would be very useful for the operation, management and design of sludge wastewater treatment systems [163].

## 9. Future Prospects and Challenges

Existing studies have effectively implemented different microalgae consortium systems for the removal of nutrients from wastewaters discharged from various sources: however, more work is needed, so that the culturing parameters can be optimized for mass scale utilization. Firstly, the sustained treatment of multiple pollutants demands an appropriate preference of the microorganisms incorporated in the consortium, because the contaminants may reduce the photosynthetic action, thus reducing the potency of the treatment. Furthermore, to achieve a more efficient consortium system, capable of degrading particulate pollutants, additional studies are needed, concerning the engineering of novel consortium systems and the pattern of artificial microbial communities, where at least one of the species included should be genetically engineered: this is because the stability of a microbial consortium is dependent on the communication (the exchange of molecular signals and metabolites) within that consortium or the individuals; therefore, the engineering of a species will allow the elimination or re-introduction of microorganisms as per the requirement, hence presenting high pollutant-removal ability. More effort is therefore still required, to overcome these challenges [45,48]:

(i)    the prolongation of homeostasis;
(ii)   maintaining the prolonged potency of the consortium, even at the time of gene transfer;
(iii)  the inclusion of stable alterations in the genomes of microbes taking part in the consortium;
(iv)   the improved performance of the consortium system.

In addition, most of the research concerning microalgae consortiums in nutrient remediation has been conducted on a lab scale that might not exemplify actual conditions; essential advances include [21,164]:

(i)    studying the influence of various environmental conditions, such as nutrient availability, light, temperature and pH, on the behavior of consortium systems;
(ii)   experimenting on a mass scale;
(iii)  gaining a complete understanding of the associations, such as commensalism, mutualism and parasitism, taking place between the microalgae, the bacteria and the fungi which, to date, have not been well described;
(iv)   evolving authentic mathematical models (such as BIO_ALGAE) that accurately represent the consortium behavior: this might be very supportive, for the determination of operational conditions and process design.

In microalgae-bacterial consortiums, in spite of the fact that specific bacterial species promote microalgae cultivation and wastewater treatment by supplying growth regulating signals and nutrients, the stability and sustainability of the consortium process is still challenged via non-targeted bacterial blooms. Concerning the work that identified bacteria either from wastewater or from the phycosphere, only a few—barely one in hundreds— were recognized as assisting microalgae growth [133,165–167]. The bloom of additional undesirable bacteria can take place with high expectations, which is regarded as "biological contamination": this is harmful for microalgae cultivation, results in frequent culture crashes, and further obstructs the commercialized evolution of microalgae biomass production, mainly in those fields which are utilizing wastewater as a medium for decreasing the cost [68,168,169]. The obstacles confronted by employing microalgae–bacterial consortiums in wastewater involve the possible negative influence of bacteria on algae, and an inadequate understanding of consortium behavior on a mass scale. The destructive consequences of bacteria on microalgae biotechnology include the following:

(i)    degrading the quality of algal biomass through consumption of valuable algal bio-products;
(ii)   directly hindering the growth of algae either by nutrient competition or by an allelopathic action;
(iii)  increasing the chances of microalgae culture contamination by pathogenic bacteria.

It remains a challenge to upgrade the implementation of microalgae–bacterial consortiums in wastewater, and to assure the desired yield and quality of algal biomass, because of the engineering and biological factors that demand assistance from mathematical modeling and process control [170]: in this regard, a mathematical model to regulate a particular high-rate algal pond has been developed successfully, whereas additional general techniques are still required to upgrade microalgae–bacterial consortium for wastewater treatment without adversely affecting biomass production, where advanced process control and algorithms cannot be missed out. Similarly, the mutual relationship between fungi and microalgae provides a novel approach to the areas of wastewater treatment, biofuel production and microalgae harvesting. However, research on microalgae–fungi co-cultivation is still small-scale, and the literature has not yet communicated the mass scale implementation of this process. There are excellent benefits to be gained from this technology, but various bottlenecks and challenges have yet to be resolved:

(i)    the preferences of microalgae and fungi species, and their co-cultivation conditions, highly affect this process; presently, filamentous fungi are ratified to be efficient in microalgae harvesting: unfortunately, most of them are pathogenic, and therefore do not have any practical application value;
(ii)   inadequate co-cultivation conditions result in reduced flocculation efficacy; the impact of different parameters on the flocculation methods of microalgae and fungi are still in an investigative phase; optimized co-cultivation parameters involving agitating, addition of carbon source and illumination demand high cost, thus hindering implementation on a mass scale;

(iii)  generally, wastewater from natural sources contains bacteria: however, most of the studies have utilized wastewater after its filtration and sterilization; it is quite difficult to construct a distinct microalgae–fungi system that totally lacks bacteria, but fungi can effectively guard microalgae from bacterial interference.

In addition, the perspectives of microalgae–fungi consortium include the following. Firstly, biological control of fungi, excluding the risk of environmental contamination and particular microalgae strains, must be tested and identified for this process, or else microalgae and fungi species must be chosen with great flexibility. The co-cultivation conditions require more optimization, and additional attention should be contributed to the parameters beneath natural light, excluding the inclusion of carbon source and the alteration of pH, thus enhancing the probability of mass-scale implementation along with economic advantages. Secondly, exploration of microalgae and fungi species at molecular level, involving proteomics or metabolomics and amino acid composition, must be performed, for searching the vital genes or proteins included in the method of bio-flocculation, in order to assist sourcing for the relationship between microalgae and fungi. In addition, the composite three-way association among bacteria, fungi and microalgae is unknown, and this interaction should be taken into consideration for practical applications. Thirdly, the benefit of microalgae–fungi consortiums in wastewater treatment is exclusively rooted in the truth that fungi can enhance the growth rate of microalgae, and its efficiency in wastewater treatment is chiefly differentiated from that of microalgae monosystems.

Nonetheless, based on the above studies, a different microalgae consortium system has been seen as an optimistic approach to wastewater remediation, along with cost-effective algal-derived biofuel applications. Recent evolutions of this biotechnology have been consequential; but there remain some challenges; therefore, control procedures to continue prolonged operation of the consortium system, in spite of alterations in environment and biological contamination, have yet to be extensively examined. Additional research efforts and data interpretation should be assigned to the significance and boosting of microalgae consortiums [171]: this is because the entire proposal is the first step on the way to applying the concept of ecological engineering in microalgae remediation methods, leading to the emergence of more efficient and resilient treatment systems.

## 10. Conclusions

In order to eliminate toxic contaminants and pollutants from the environment, microalgae consortiums are one of the best of the reported approaches. Interactions among algae and other microorganisms are complicated, and the utilization of microalgae consortiums in this area is still in the developing phase, basically because of the extensive variety of practicable combinations that can be achieved. In addition, very little has been investigated about the relationships initiated between photosynthetic microorganisms. Existing studies have stated the potency of microalgae–bacterial and microalgae–fungi consortiums, as compared to algae–algae consortiums, because they can be utilized as a substitute for both the tertiary and secondary treatment steps involved in wastewater treatment, whereas microalgae consortiums can only be implemented in wastewater polishing (as a substitute of the tertiary treatment step); however, only a few of the studies have reported on the screening of particular symbiotic strains and the development of a specific and well-constructed symbiotic system. Due to the complication of microorganisms in co-culture systems developed by mixed flora, the reliability of the system is hard to monitor, and eventually influences the effect of wastewater treatment: this is not favorable to the study of the interaction mechanisms involved between microalgae and symbiotics; however, this consortium system demands additional research and effort, so that this novel technology can be practiced and commercialized on an industrial scale, for a more prosperous and liveable society.

**Author Contributions:** Conceptualization; Data curation; Writing—review and editing, P.G.; Data curation; Writing—original draft, P.B.; Conceptualization; Supervision, V.K.; Data curation; Investi-

gation; Writing—review and editing, M.S.V.; Data curation; Proof Reading, A.V.G. All authors have read and agreed to the published version of the manuscript.

**Funding:** RUDN University scientific projects grant system, project No: 2027042000.

**Institutional Review Board Statement:** Not applicable for studies not involving humans or animals.

**Informed Consent Statement:** Not applicable.

**Data availability statement:** Not applicable.

**Acknowledgments:** This publication has been supported by the RUDN University scientific projects grant system, project No: 2027042000.

**Conflicts of Interest:** The authors declare no conflict of interest.

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
