# Peer review of "Algal Consortiums: A Novel and Integrated Approach for Wastewater Treatment"

_water, doi:10.3390/w14223784_

Round 1
Reviewer 1 Report
The manuscript entitled ‘Algal consortium: A sustainable and integrated approach for 2 wastewater treatment' is a review paper that aims to provide knowledge of the symbiotic relationships established between microalgae and other microorganisms in wastewater treatments including the mechanism responsible behind removal of various contaminants and pollutants from wastewater. The paper further addresses the challenges that are restricting large-scale implementation of these consortiums.
I agree that this work can indeed provide a deep understanding of the symbiotic relationships established between microalgae and other microorganisms in wastewater treatments.
Overall, this is a good study, with useful information achieved; worthy for publication.
Consequently, this work could be accepted by Water after a major revision. The major revision should be focused on the following:
1. Line 264-265: ‘On that account, some of the existing studies have communicated the benefits of utilizing microalgae consortia above single-species cultures’.
Can you give more citations? And also, please write the single-species cultures.
2. Line 270: ‘Such consortiums can either occur naturally in the environment’. Can you be more specific? In which environments can be found such consortiums?
3. In subsections, 4.1, 4.2 and 4.3, please give more information about the genus and name of the microorganisms.
4. All figures quality is poor and needs to be enhanced.
5. Conclusions. Conclusion is too long; you begin introducing concepts that you have already covered earlier in the document. A conclusion is typically short.
6. References missing. I could not check

Author Response
Response to Reviewer 1 is attached in separate word file

Author Response
Response to Reviewer 2 is attached in separate word file

Reviewer 3 Report
The review manuscript entitled referenced above addresses and important and emerging area in waste water management. The topic timely and the organization of the manuscript is appropriate.
However, the manuscript is deficient in some critical areas than preclude me from making a judgement on acceptability for publication.
The first issue is missing sections. The downloaded copy of the manuscript skips from line 132 to 176 with the text in-between missing entirely. Second, the references cited section is also missing, so it is impossible to determine which papers are being cited.
Finally, I find the writing style to be unnecessarily verbose with too many sentences beginning in parenthetical phrases. Many sentences are also quite vague and uninterpretable such as the sentence on lines 189-192, 198-199. That entire paragraph is difficult to interpret, poorly organized, and occasionally misleading or simplistic. The discussion of "low light" inhibition for example has no context and is therefor meaningless.
I would be willing to have another look at a revised manuscript that is complete and substantially re-written in simple direct sentences with more care taken with word choice. Example: line 45, do you really mean "prior" at the end of that sentence? Wouldn't "primary" be a better choice?
Author Response
Response to Reviewer 3 is attached in separate word file

Reviewer 4 Report
The content of your submitted review paper is timely. Your review paper will contribute the development of economical and sustainable wastewater treatment using algal consortium. But, there are many unclear points in your review paper. I have judged major revision is required before accepting your review paper. The followings are the weak points in your review paper.
1. Your paper is redundant and difficult to read. One sentence is too long. Please try to write your review paper in briefly.
2. There is no title of Fig.1.
3. Fig. 3 You must show the flow direction in each system. As for the closed system, lighting system is important. Please show the lighting system in the figure.
4. Table 1 Cultivation condition → Preculture and cultivation using wastewater is mixed in this Table. Please write the cultivation condition as S.no. 1. I suppose every cultivations were conducted in batch culture. Initial algal concentration and the cultivation time are required. For the cultivation of algae, initial substrate concentration is important. Please show the initial substrate concentration.
5. Table 1, S.no. 6 → activated sludge ? Is this secondary effluent?
6. Table 1 P-PO4 Total phosphate → ortho-phosphate
7. Table 2 You must show the way of cultivation(batch or continuous culture?), reactor type and cultural condition(loading rate, lighting time, pH, sludge concentration in this Table).
8. For the proper operation of algal consortium to the wastewater treatment, solid-liquid separation is required. In this sense, flocculation of algal consortium is important. Please discuss the settling velocity and sludge volume index like SV30 for algal floc, which is the essential information for the design of settling tank.
9. Information on the yield coefficient is an important for the design of excess sludge processing plant. Please describe these information in your review paper.
10. For actual application of algal system to the wastewater treatment, mixing, lighting at night time, depth of algal tank, recycle ratio of settled algal sludge etc. are required information. Please discuss these points in your review paper.
11. More invention is required to Figure 4 and 5. Your illustration is too ordinary.
12. Please unify NH3-N and NH4-N.
Author Response
Response to Reviewer 4 is attached in separate word file

Round 2
Reviewer 1 Report
Accept in present form
Reviewer 4 Report
You have revised your submitted paper properly by taking into accounts my comments.I have judged your revised paper reached to the acceptance level of Water.